# Ethanol-Induced Cell Damage Can Result in the Development of Oral Tumors

**DOI:** 10.3390/cancers13153846

**Published:** 2021-07-30

**Authors:** Lore Hoes, Rüveyda Dok, Kevin J. Verstrepen, Sandra Nuyts

**Affiliations:** 1Laboratory for Systems Biology, VIB-KU Leuven Center for Microbiology, 3000 Leuven, Belgium; lore.hoes@kuleuven.be (L.H.); kevin.verstrepen@kuleuven.be (K.J.V.); 2Laboratory of Genetics and Genomics, Centre for Microbial and Plant Genetics, KU Leuven, 3000 Leuven, Belgium; 3Laboratory of Experimental Radiotherapy, Department of Oncology, KU Leuven, 3000 Leuven, Belgium; ruveyda.dok@kuleuven.be; 4Department of Radiation Oncology, Leuven Cancer Institute, University Hospital Leuven, 3000 Leuven, Belgium

**Keywords:** oral potentially malignant disorders, oral squamous cell carcinoma, ethanol, molecular alterations, mutational signatures

## Abstract

**Simple Summary:**

Alcohol consumption is linked to 26.4% of all lip and oral cavity cancer cases worldwide. Despite this clear causal relationship, the exact molecular mechanisms by which ethanol damages cells are still under investigation. It is well-established that the metabolism of ethanol plays an important role. Ethanol metabolism yields reactive metabolites that can directly damage the DNA. If the damage is repaired incorrectly, mutations can be fixed in the DNA sequence. Whenever mutations affect key regulatory genes, for instance cell cycle regulating genes, uncontrolled cell growth can be the consequence. Recently, global patterns of mutations have been identified. These so-called mutational signatures represent a fingerprint of the different mutational processes over time. Interestingly, there were ethanol-related signatures discovered that did not associate with ethanol metabolism. This finding highlights there might be other molecular effects of ethanol that are yet to be discovered.

**Abstract:**

Alcohol consumption is an underestimated risk factor for the development of precancerous lesions in the oral cavity. Although alcohol is a well-accepted recreational drug, 26.4% of all lip and oral cavity cancers worldwide are related to heavy drinking. Molecular mechanisms underlying this carcinogenic effect of ethanol are still under investigation. An important damaging effect comes from the first metabolite of ethanol, being acetaldehyde. Concentrations of acetaldehyde detected in the oral cavity are relatively high due to the metabolization of ethanol by oral microbes. Acetaldehyde can directly damage the DNA by the formation of mutagenic DNA adducts and interstrand crosslinks. Additionally, ethanol is known to affect epigenetic methylation and acetylation patterns, which are important regulators of gene expression. Ethanol-induced hypomethylation can activate the expression of oncogenes which subsequently can result in malignant transformation. The recent identification of ethanol-related mutational signatures emphasizes the role of acetaldehyde in alcohol-associated carcinogenesis. However, not all signatures associated with alcohol intake also relate to acetaldehyde. This finding highlights that there might be other effects of ethanol yet to be discovered.

## 1. Alcohol Consumption and Its Adverse Effects Know a Long History

The first traces of alcohol fermentation already date back from 7000 before Christ [1]. Throughout the early modern period, consumption of alcohol vastly increased. In the 1500s, average alcohol consumption reached a hundred liters per person per year in Spain and Poland. The English population consumed on average seventeen pints (beer or ale) per week in contrast to an average of three pints today [2]. With this increasing consumption of alcoholic beverages, physicians started to notice the adverse effects of alcohol. Already in the eleventh century, a physician in Constantinople reported liver inflammation due to excessive wine drinking [2,3]. However, it was only until the late eighteenth century that alcohol addiction and abuse were acknowledged as physical and mental health problems [4]. In the early twentieth century, professor R. Pearl published his research about alcohol and longevity. He found that heavy alcohol consumption was associated with higher mortality rates [5,6]. Thereafter, research on how alcohol consumption is related to several medical conditions gained momentum which contributed substantially to our current knowledge on the health effects of ethanol [6].

Evidently, heavy alcohol consumption negatively impacts human health. Alcohol damages nearly every organ in the human body. In fact, more than 60 different diseases and conditions have been causally linked to alcohol consumption [6,7,8]. In 2016, the World Health Organization (WHO) estimated that 3 million deaths worldwide were attributable to alcohol consumption [7,8]. Globally, the use of alcohol was ranked as the seventh leading risk factor for both deaths and disability-adjusted life-years in 2016 [7]. Alcohol use is mostly associated with non-communicable diseases including malignant neoplasms (cancer), diabetes mellitus, alcohol use disorders, cardiovascular diseases, and alcohol liver diseases [9,10]. Specifically for cancer, Praud et al. quantified that in 2012 770,000 cancer cases were associated with alcohol consumption. This roughly corresponded to 5.5% of all cancer cases worldwide [11]. This estimate was recently updated in the population-based study published by Rumgay et al. in The Lancet Oncology. In 2020, 741,300 or 4.1% of all new cancer cases were estimated to be attributable to alcohol consumption [12]. In 2012, the International Agency for Research on Cancer (IARC) formally classified alcoholic beverages and acetaldehyde, the first metabolite of ethanol, as type I carcinogens to humans [13].

Notably, alcohol consumption has pleiotropic effects on human health and these are strongly dose-dependent. It has been suggested that low alcohol consumption has beneficial cardiovascular effects. For instance, a J-shaped relationship between alcohol intake and ischaemic heart disease was observed in the Global Burden of Disease Study in 2016 [7]. Although interesting, the topic of this review paper is the relationship of ethanol with oral carcinogenesis.

## 2. Oral Cancers Are Still Frequently Diagnosed despite Avoidable Risk Factors

Oral cancers are a defined subset of head and neck cancers. These include cancers of the tongue, the floor of the mouth, cheeks, palate, lips, or gums but exclude cancers of the pharynx and larynx, corresponding to the International Classification of Diseases, 10th revision codes C00-C06 [14]. These oral cancers almost exclusively consist of oral squamous cell carcinomas (OSCCs) [15]. Additionally, there is a gender discrepancy as oral cancers are particularly common in men [15,16]. A possible reason for this is that men are generally more exposed to risk factors as discussed in the next paragraph. In 2018, 354,864 new patients were diagnosed with lip or oral cavity cancers corresponding to 2% of the total amount of new cancer cases [17]. Many of these cases occur in low- and middle-income countries, e.g., India, Pakistan, or Tanzania, making it the fourth most common type of cancer in these countries [14]. Despite advances in treatment, the overall five-year survival rate for oral cancers is only 50–55% [18]. Oral carcinogenesis is a complex multistep process that takes place over many years [16,19]. Genetic alterations of squamous cells can lead to uncontrolled cell growth, a hallmark of cancer. There exist multiple oral potentially malignant disorders (OPMDs) that can precede oral cancer [19]. Frequently diagnosed lesions include oral leukoplakia, which are visible as white oral patches [18,20]. Other, less common, OPMDs include erythroplakia, oral submucous fibrosis, and oral lichen planus [21]. The prevalence of these lesions increases with age and early detection is crucial to avoid malignant transformation [18,22,23].

Established risk factors for developing oral malignancies include tobacco smoking or chewing, heavy alcohol drinking, and human papillomavirus (HPV) infection [18]. Firstly, tobacco is an undeniable risk factor for several cancer types, including oral cancers. Epidemiological studies associate both smoked and smokeless tobacco with the formation of tumors in the oral cavity [24,25,26,27,28]. Smokeless tobacco, for instance, enhances the incidence of oral cancers almost fourfold, based on results from 36 independent studies [28]. It should be noted however that this association is dependent on geographical location, as not all studies conducted in Europa and North-America found a significant association between smokeless tobacco and increased risk of oral tumors [27,28]. Secondly, tobacco use is often accompanied by alcohol consumption. Tobacco and alcohol have a major synergistic effect on the development of oral cancerous lesions [29,30,31,32]. The increase of the odds ratio depends on the dose of both alcohol and tobacco [31]. Together, tobacco and alcohol use account for approximately 75% of oral cancers [33]. Despite the synergistic effect of tobacco combined with alcohol, also alcohol on its own is a considerable risk factor for oral cancers [34]. The role of alcohol in oral carcinogenesis will be elaborated in the next section. Men tend to drink and smoke more compared to women, which might be the reason why oral cancers are so common in men [35,36]. Lastly, infection with HPV, especially high-risk HPV16 or HPV18, is a well-recognized risk factor for oropharyngeal cancers [37,38]. The prevalence of HPV infections in oral cancers is also significant in some geographic locations, e.g., around 36% in India and Japan [39]. In contrast, only 2.2% of oral tumors were positive for HPV in a Dutch cohort [40]. HPV^+^ tumors show a distinctly different molecular and mutational landscape compared to HPV^−^ tumors [38]. Overall, HPV positivity is associated with a better prognosis [39,41].

## 3. Alcohol Is an Independent Risk Factor for Oral Carcinogenesis

Although alcoholic beverages consist of multiple components, this review will solely focus on ethanol. Quantification of the amount of ethanol consumed is critical to assess the effects of this compound on the incidence of specific tumors. Different studies use different measures and thresholds to define low, moderate, and heavy drinking. For instance, the definition of a ‘standard drink’, which originated from governmental guidelines, is highly variable between countries. In Belgium and France, one standard drink contains 10 g of pure ethanol. In contrast, a standard drink holds 14 g and even 20 g of ethanol in the US and Austria respectively [42,43]. Therefore, the comparison of different studies is not always straightforward. In this review, the unit used to compare studies is always grams of pure ethanol.

### 3.1. Epidemiological Data Indicate a Strong Correlation between Alcohol and Oral Malignancies

Long-term ethanol exposure results in the formation of tumors all over the human body. Clear causal relationships have been identified for (at least) seven different types of cancer. The strongest associations were observed for cancers originating in the oral cavity, pharynx, larynx, and esophagus [44,45,46,47,48,49]. Specifically for oral cancers, alcohol consumption is a substantial risk factor with a clear dose-response relationship [50,51,52,53,54,55]. Studies from various geographical areas show elevated odds ratios for oral cancers in combination with ethanol intake [56,57,58,59,60,61,62]. For instance, consumption of more than 40 g of pure ethanol daily, was shown to result in a threefold increased risk for developing oral tumors in Indian men [63]. A Spanish case-control study also confirmed elevated odds ratios of oral cancer associated with ethanol. Heavy drinking, defined as >50 g of ethanol a day, resulted in an odds ratio of 5.04 [64]. It should be noted however that this study also included oropharyngeal cancers when defining oral cancers. Recently, Griswold et al. performed an extensive meta-analysis, combining data from almost 600 studies, concerning the role of alcohol in disease. This analysis shows that the relative risk for developing mouth and lip cancers in men increases linearly with the amount of alcohol consumed [7]. This result also highlights that there is no ‘safe’ level of drinking when the incidence of oral tumors is evaluated. Noteworthy, lip cancer is often grouped together with oral cavity cancers. However, the main risk factors for developing (outer) lip cancer are tobacco and UV exposure [65,66]. Also, data from the International Head and Neck Cancer Epidemiology consortium confirmed that alcohol is an independent risk factor for cancers of the oral cavity, oropharynx, hypopharynx, and larynx [55]. Odds ratios for cancers of the oral cavity increased linearly with the number of drinks a day. The curve however flattened when more than 5 drinks (i.e., 60 g of pure ethanol) were consumed daily. Interestingly, the bivariate spline models suggested that the duration of drinking (measured in years) does not affect the odds ratio for oral cavity cancers [55]. In other words, the risk of developing oral cavity tumors does not increase when patients drink the same amount of alcohol for a longer time. Noticeably, some studies did not find significant associations between ethanol and the incidence of oral tumors [49,67,68,69]. Sometimes, only increased odds for oral cancers can be found when patients do not only drink alcohol but also smoke tobacco. Indeed, ethanol is believed to increase membrane permeability of the oral mucous layer which subsequently leads to increased exposure to tobacco carcinogens [70]. Although multiple studies (cited above) showed a linearly increasing relationship between alcohol and oral tumor incidence, the role of low alcohol consumption in oral carcinogenesis remains controversial. For instance, results of a cohort study conducted in the Netherlands showed that drinking less than 15 g of ethanol per day did not significantly increase the risk for developing oral cavity cancer [60].

In addition to associations of oral carcinomas with ethanol intake, OPMDs have also been investigated. A large prospective study evaluated the link between ethanol exposure and occurrence of OPMDs in men [71]. Lesions taken into account in this study include leukoplakia, erythroplakia, erythroleukoplakia, oral lichen planus, and oral epithelial dysplasia. The multivariate relative risk for developing oral lesions was 2.5 for men who drank more than 15 g of pure ethanol daily (which roughly corresponds to 380 mL of a light beer containing 5% ethanol) [71]. A case-control study from Puerto Rico did however not find a significant positive association between the risk for developing OPMDs and ethanol exposure [72]. Similarly, cross-sectional studies conducted in India and Cambodia did not find a significant influence of alcohol drinking on the incidence of OPMDs [73,74]. Recently, however, several meta-analyses evaluated the malignant transformation risk for oral lichen planus, which is a chronic inflammation of the oral mucous membranes. While the overall malignant transformation rate is low (∼1%), alcohol consumption significantly increases the risk for malignancies emerging from oral lichen planus [75,76,77].

The studies above are mostly using classical or observational epidemiology to investigate the potential carcinogenic role of ethanol. Since the last decade, however, genetic epidemiology is gaining popularity. In this field, Mendelian randomization (MR) is used to investigate causal exposure-disease interactions. MR employs the random assignment of genes during gamete formation to reduce confounding, which was extensively reviewed by Smith et al. [78]. In short, the first step is to identify genetic variants which are directly associated with the exposure. The next step for MR is to divide the study population into two groups based on these specific polymorphisms linked to the exposure. Lastly, the incidence/risk of the disease of interest is investigated in both groups to assess if the exposure is indeed causally linked to the disease [78]. Considering ethanol, there are known polymorphisms in ethanol-metabolizing genes which alter exposure to ethanol [78,79]. For instance, carrying the *ADH1B* rs1229984 polymorphism was associated with less ethanol intake [80]. Recently, many more variants related to alcohol use have been identified [81]. Using MR, Gormley et al. confirmed that both alcohol and tobacco are independent risk factors for oral and oropharyngeal cancers with inverse variance weighted odds ratios of 2.1 and 2.6 respectively [82].

### 3.2. In Vivo Data Support a Causal Effect of Alcohol on Oral Tumor Incidence

It is important to acknowledge that association studies do not necessarily imply a causal relationship between ethanol and oral cancers. It is well-known that confounders or reverse causality can result in spurious associations [78]. Therefore, multiple in vivo studies have been conducted as well to assess the carcinogenic effect of ethanol. Often, the consequence of ethanol exposure is evaluated in chemically-induced cancer models. This way, it was observed that 8% ethanol administration in addition to 4-nitroquinoline-1-oxide, a toxic quinoline which induces oral tumors [83], promotes malignant transformation in C57BL/6J mice [84]. Also in male Fischer-344 rats, 7% ethanol administration on top of N′-nitrosonornicotine or 4-(methylnitrosamino)-1-(3-pyridyl)-1-butanone increased the incidence of oral tumors two-fold [85]. Importantly, Soffritti et al. also performed an in vivo study without an additional carcinogenic chemical. In their experiment, Sprague-Dawley rats were exposed to 10% ethanol *ad libitum* for about 3 years (until spontaneous death). Ethanol administrated rats developed tumors at various sites, including malignant tongue, lip, and oral cavity tumors [86].

Additionally, rabbits solely exposed to ethanol for 12 months developed leukoplakia-like epithelial dysplasia, which is a common OPMD that can be a progenitor stage of squamous cell carcinomas [87,88]. Also more dysplasia was detected in tongue and pharynx tissues of Wistar rats exposed to 30% ethanol *ad libitum* for 260 days [89].

Based on these in vivo and epidemiology data, the IARC concluded there is indeed a causal relationship between ethanol exposure and oral carcinogenesis. Interestingly, however, the exact molecular mechanisms whereby ethanol can induce OPMDs or tumor formation are still not fully understood. In the following sections of this review, we will address precisely this question.

## 4. Various Molecular Alterations Have Been Attributed to Ethanol

### 4.1. Acetaldehyde Can Accumulate in the Oral Cavity

Ethanol is known to damage eukaryotic cells in different ways. Mostly, its carcinogenic effects are assigned to the intermediary metabolites, acetaldehyde and reactive oxygen species (ROS), formed during oxidative ethanol metabolism. Multiple enzymes are involved in the conversion of ethanol. A concise overview of ethanol metabolism and genes involved is given in Figure 1. Alcohol dehydrogenases (ADHs) convert a considerable part of the ethanol into acetaldehyde. Additionally, cytochrome P450 enzymes (CYP), mostly CYP2E1, can oxidize ethanol into acetaldehyde as part of the microsomal ethanol oxidizing system (MEOS). Thereafter, aldehyde dehydrogenases (ALDHs) further metabolize acetaldehyde which yields acetate. ROS can either be generated directly, during the conversion of ethanol into acetaldehyde by CYP2E1, or indirectly, when NADH is re-oxidized to NAD^+^ in the mitochondria [90,91]. Although the liver is the primary organ which metabolizes ethanol, expression of *ADH*, *CYP* and *ALDH* genes have also been detected in the oral mucosa [92,93,94].

In the oral cavity, there is another important factor that contributes to the accumulation of acetaldehyde: the oral microbiome. Several bacterial species that belong to *Streptococcus* and *Neisseria* genera, commonly detected in the oral microbiome, have been shown to produce acetaldehyde from ethanol [95,96]. For instance, *Neisseria mucosa*, *Neisseria flavescens* and *Streptococcus mitis* were shown to produce 272.8 ± 65.5, 168.4 ± 8.6 and 90.2 ± 31.3 μM acetaldehyde respectively, in the presence of 11 mM ethanol and 100 mM glucose [97]. Homann et al. (1997) established that acetaldehyde levels in the saliva of human volunteers can reach ∼ 140µM after a moderate dose of ethanol (0.5 g of ethanol/kg) which is 10–100 fold higher compared to blood acetaldehyde levels (∼1–5 µM) [46,98]. Alcohol consumption in combination with smoking can even increase the salivary acetaldehyde up to 400 µM [99]. This concentration easily reaches the carcinogenic threshold, defined as acetaldehyde levels above 50–100 µM [100,101]. Interestingly, the composition of the oral microbiome can also be altered towards more acetaldehyde producing bacteria by heavy alcohol consumption and tobacco smoking [95,102,103].

A last source of acetaldehyde in the oral cavity is the alcoholic beverage itself. Lachenmeier and Sohnius chemically analyzed more than 1500 beverages and quantified acetaldehyde concentrations using gas chromatography. They concluded that beer, wine, and spirits contain on average 0.204 mM, 0.765 mM, and 1.48 mM acetaldehyde respectively [100].

### 4.2. Metabolites of Ethanol Can Directly Affect the DNA by Formation of Adducts and Crosslinks

#### 4.2.1. Acetaldehyde-Derived DNA Adducts and Crosslinks

Acetaldehyde can directly interact with the DNA base pairs resulting in DNA adducts (Figure 2). The first and most abundant adduct is N^2^-ethyl-2′-deoxyguanosine (N^2^-EtdG) [114,115]. N^2^-EtdG originates from N^2^-ethylidene-2′-deoxyguanosine (N^2^-EtidG) which is produced via direct interaction of a single acetaldehyde molecule with deoxyguanosine (dG). However, this Schiff base is unstable in vivo (half-life of 24 h at 37 °C), but reduction by e.g., vitamin C or glutathione yields the stable N^2^-EtdG adduct [114,115]. In vitro experiments showed that N^2^-EtdG can block the replicative DNA polymerase α [116] in contrast to DNA polymerase δ which was not significantly blocked by this small adduct [117]. Upon blockage of a replicative polymerase, this lesion can be efficiently bypassed by translesion DNA polymerase η [116,117]. In vivo data are in agreement with these earlier findings. Translesion polymerases are probably the most important way to remove this lesion, as BER or direct repair have not been shown to repair N^2^-EtdG [101]. Resolution of the N^2^-EtdG block by translesion DNA polymerases mostly results in frameshift mutations that are weakly mutagenic [101,118]. Extrapolating the consequences seen for N^2^-EtdG to N^2^-EtidG should however be done with caution as these adducts interact differently with deoxycytidine (dC). N^2^-EtdG forms three hydrogen bonds with dC while N^2^-EtidG can only form two, which renders a G:C base pair that is only as stable as an A:T base pair [101].

A second important adduct is α-methyl-γ-hydroxy-1,N^2^-propano-2′-deoxyguanosine, also referred to as crotonaldehyde-derived N2-propanodeoxyguanosine (Cr-PdG) [46,119]. For the formation of Cr-PdGs, two molecules of acetaldehyde are necessary as was unequivocally shown by Garcia et al. [120]. In physiologically relevant conditions (100 μM acetaldehyde), polyamines probably convert two acetaldehydes into crotonaldehyde which in turn can interact with dG forming Cr-PdG [119]. Although less abundant, Cr-PdG adducts are more mutagenic compared to N^2^-EtdG. Cr-PdG adducts can exist in a ring-opened or a ring-closed confirmation. In a ring-closed state, Cr-PdG inhibits base pairing with dC which could result in replication blocking [121]. In a ring-open state however, a reactive aldehyde group is exposed which can interact with proteins or dG resulting in DNA-protein and DNA-DNA interstrand and possibly also intrastrand crosslinks [121]. Multiple repair pathways have been shown to be involved in Cr-PdG repair, including nucleotide excision repair and translesion polymerase synthesis [122,123]. Interstrand crosslinks (ICLs) are primarily repaired by the Fanconi anemia (FA) pathway. Acetaldehyde hypersensitivity was detected in FA-deficient DT40 chicken B-cells, mice and DLD1 human cells which highlights the importance of the FA repair pathway for acetaldehyde-induced damage [124,125,126]. The FA repair is initiated when the FANCM–FAAP24–MHF1/2 complex recognizes the stalled replication fork due to an ICL. Subsequently, unhooking of the ICLs takes place by specific nucleases and this converts the stalled replication fork into a double-strand break (DSB), on one end, and unhooked ICL, on the other end. Translesion DNA synthesis can bypass the unhooked crosslinked nucleotide to restore the nascent DNA strand. To repair the DSB, mostly homologous recombination is used [127]. The FA repair pathway always introduces a DSB which can induce large genomic instability when it gets exposed. However recently, Hodskinson et al. unexpectedly discovered a new and faster pathway to repair ICLs [128]. This new route also requires replication fork convergence but does not cut the DNA which yields a safer fix for ICLs [128]. This finding might indicate that acetaldehyde induces less DSBs in vivo than was initially assumed. Nevertheless, acetaldehyde induced DNA crosslinks can still be mutagenic as the repair still requires error-prone polymerases such as REV1 and pol ζ [128].

A third DNA adduct that was detected after acetaldehyde exposure is 1,N^2^-etheno-2′-deoxyguanosine (NϵdG). This adduct is suggested to result from lipid peroxidation and not a direct interaction between acetaldehyde and the DNA. More specifically, acetaldehyde can induce lipid peroxidation in which epoxidized α,β-unsaturated aldehydes are formed that can subsequently interact with dG which results in NϵdG [120,121,129]. Garcia et al. showed that exposure of IMR-90 primary human lung fibroblasts to 155 μM for 3 h significantly increased NϵdG adduct formation [120]. This concentration can easily be reached in the saliva, as discussed before. The mutagenic consequences of NϵdG are not fully understood yet. In vitro, this adduct blocks the replicative DNA polymerase δ and bypass of this lesion by error-prone polymerases is needed which has varying mutagenic consequences [130]. However, in vivo data hypothesized that NϵdG leads to replication fork collapse and generation of DSBs, but this was not yet confirmed by other studies [121]. How this adduct is repaired is still uncertain. It was previously thought that BER is the main route for repair, but recently Thelen et al. showed that alkyladenine DNA glycosylase cannot use NϵdG as a substrate [131].

In adult Rhesus monkeys exposed to ethanol, N^2^-EtdG levels were 2.8 fold increased, as analyzed by LC-ESI-MS/MS [132]. The animals consumed ethanol (5% in water) *ad libitum* for 12 months after which mucosal tissues were isolated and analyzed [132]. The researchers estimated that the monkeys consumed on average 2.3 ± 0.8 g/kg of ethanol per day, which is equivalent to nine drinks containing 15 g ethanol each. In *ALDH2*-knockout mice, known to accumulate acetaldehyde, drinking of 5% ethanol for 8 weeks significantly increased esophageal N^2^-EtidG levels (9.73 ± 2.33 adducts/10^7^ bases) [133]. In addition to animal studies, acetaldehyde-derived DNA adducts have also been quantified in human individuals. For instance, increased levels of N^2^-EtdG were reported in lymphocytes of alcoholic patients (consumed > 50 drinks per week) [114]. Interestingly, Balbo et al. examined the kinetics of N^2^-EtdG formation in oral cells of human volunteers. They observed that N^2^-EtdG levels increased drastically (up to 100-fold) within 4 h after administration of low doses of ethanol [134]. Despite a lower abundance, also Cr-PdG adducts have been observed in the DNA of human subjects. In Japanese alcoholic patients, the level of Cr-PdG adducts in blood DNA was significantly increased in patients that carried the *ALDH2*2* or *ALDH2* rs671 allele [135]. This variant of *ALDH2* renders an inactive enzyme and therefore individuals carrying this allele build up more acetaldehyde.

#### 4.2.2. ROS-Derived DNA Adducts

ROS are always present during normal cellular metabolism, but drinking ethanol can enhance cellular ROS levels. Ethanol metabolism by CYP2E1 is predominantly causing ROS levels to rise. *CYP2E1* is highly expressed in the liver making this organ highly susceptible to ethanol-induced oxidative damage [136]. But also in oral carcinogenesis, oxidative stress is an important contributor [137]. In addition to the direct formation of ROS during ethanol metabolism, abuse of alcohol can also deplete key radical scavengers such as glutathione, vitamin C, and vitamin E [138,139]. Mostly, vitamin depletion is linked with malnutrition in alcohol-dependent patients. Notably, also these indirect effects of alcohol abuse on oxidative stress have been mostly studied in the liver.

ROS can result in direct oxidation of DNA base pairs and possibly even in DNA-DNA crosslinks (reviewed by [140]). An abundant lesion is 8-hydroxy-2′-deoxyguanosine (8-OHdG) which results from the interaction of OH^•^ with dG (Figure 3A) [141]. Keto-enol tautomerism of 8-OHdG favors the oxidized product 8-oxo-7,8-dihydro-2′-deoxyguanosine (8-oxodG). In literature, both 8-OHdG and 8-oxodG refer to the same compound and this is widely used as a biomarker for oxidative stress and carcinogenesis [141]. This lesion is also mutagenic because 8-oxodG mispairs with A which results in frequent GC →TA transversions [142]. Oxidized base pairs are often efficiently removed by base excision repair (BER) [140]. Interestingly, however, ethanol can indirectly inhibit 8-oxo-guanine-DNA- glycosylase 1, the primary DNA glycosylase that removes 8-oxodG, due to induction of nitric oxide [143,144].

Ethanol-induced ROS can interact with lipid molecules present in the cell membrane which results in lipid peroxidation [145]. Lipid peroxidation generates malondialdehyde (MDA) and 4-hydroxy-2-nonenal (4-HNE) [145]. MDA originates from nonenzymatic lipid peroxidation of unsaturated fatty acids and the 4-HNE from the oxidation of long-chain polyunsaturated fatty acids [146]. These products can interact with the DNA thereby forming exocyclic etheno-adducts such as 1,N^6^-etheno-2′-deoxyadenine (NϵdA) and 3,N^4^-etheno-2′-deoxycytidine (NϵdC) (Figure 3B) [147]. These etheno-adducts are generally repaired by BER. It is well known that NϵdA is a substrate of the alkyladenine DNA glycosylase [148]. NϵdC on the other hand is repaired by human glycosylases SMUG1 and TDG [149,150]. If not repaired, these adducts are strongly mutagenic as they can induce multiple types of base-pair substitutions [151]. The biological relevance of 4-HNE DNA adducts is also emphasized by the finding that this was found to mutate codon 249 of human *TP53* (which encodes p53) thereby giving the cells a growth advantage [152]. Notably, MDA and 4-HNE are also known to react with certain amino acids which results in protein adducts [153,154]. These protein adducts can be immunogenic thereby triggering tissue inflammation [146]. Tissue inflammation is especially linked to alcohol-induced liver damage, but can also play a role in oral carcinogenesis [146,155].

Gingival tissue of male Wistar rats exposed to ethanol for instance showed a significant increase of 8-OHdG and decrease of glutathione levels [139]. In Sprague-Dawley rats that chronically consumed ethanol, elevated levels of 8-oxodG were also detected in the liver and even in the esophagus, when rats received a vitamin-depleted diet [156]. Additionally, increased levels of MDA were observed in parotid and submandibular salivary glands of female Wistar rats when exposed to ethanol [157]. This study attempted to mimic binge drinking by feeding the rats an ethanol dose of 3 g/kg/day and this for 3 days a week [157]. Also, 4-HNE levels were found to be significantly increased in tongue tissues of female mice exposed to 20% of ethanol in their drinking water for 15 weeks [158]. In human patients diagnosed with OSCC or OPMDs, including leukoplakia, oral lichen planus, and submucous fibrosis, levels of 8-OHdG and MDA were significantly increased in comparison to healthy individuals [159]. Also in a small subset of British and Japanese patients reporting alcohol misuse, protein adducts with acetaldehyde, MDA and 4-HNE were found in oral biopsies [160]. The staining of MDA-protein adducts was also significantly correlated with acetaldehyde and *CYP2E1* expression [160]. In liver samples from patients with a history of alcohol abuse, the NϵdA adduct was detected and it was later established that the presence of this adduct correlates with *CYP2E1* expression [161,162]. Interestingly, a significant increase of NϵdA was also observed in vitro in *CYP2E1* overexpressing HepG2 cells when exposed to 5–25 mM ethanol [162]. These findings were later also confirmed in esophageal tissue samples [163].

### 4.3. Ethanol Exposure Alters the Epigenome

It is very well known that global epigenetic alterations are a hallmark of cancer development [164]. Increasing evidence suggest that ethanol-induced tumorigenesis can also (partially) be explained by epigenetic modifications. Notably, most data have come from research in the liver and the brain [165]. However, these mechanisms might be universal and therefore also affect cells in the oral cavity.

#### 4.3.1. Ethanol Leads to DNA Hypomethylation

DNA methylation patterns are important transcriptional controls of gene expression [166]. DNA methylation occurs almost exclusively on carbon 5 of cytosine nucleotides. In a sequence context, often cytosines that precede guanines, so-called CpG dinucleotides, are the target for methylation. CpG clusters or islands are abundant in promoter regions or regions that contain repetitive DNA sequences [166]. Mostly, methylation in a promoter region is associated with gene silencing. Enzymes that are involved in maintaining methylation patterns are DNA methyltransferases and the most important methyl donor is S-adenosyl-L-methionine (SAM) [166].

Persuasive evidence exists that ethanol can disturb DNA methylation patterns by affecting the one-carbon metabolism, a pathway in which a chemical unit containing one carbon atom (e.g., a methyl group) is transferred from a donor to an acceptor [166]. Ethanol can directly affect key enzymes of the one-carbon metabolism, including methionine synthase (MTR), methionine adenosyl transferase (MAT), and DNA methyltransferase (DNMT). Ethanol can inhibit MTR and MAT activity which results in reduced SAM levels [167,168,169]. Indeed in rats fed alcohol for nine weeks, hepatic levels of SAM and DNA methylation fell by about 40% [170]. DNMT enzyme activity can be inhibited by ethanol and acetaldehyde as well which results in altered DNA methylation patterns [171,172].

Additionally, folate uptake and metabolism are altered in heavy drinkers. In the form of 5-methyltetrahydrofolate, this molecule is involved in the remethylation of homocysteine to methionine, a precursor of SAM [173]. Reduced folate levels have been reported in alcohol abusers which can be due to decreased dietary intake, decreased absorption, or increased urinary excretion [174]. Folate deficiency can reduce SAM levels which consequently affects the DNA methylation capacity of the cells [166,173]. It should however be noted that folate also plays a role in other essential cellular pathways and therefore the link between folate deficiency and malignant transformation is not straightforward [174].

Global DNA hypomethylation has been detected in cancer types with a strong ethanol etiology. Smith et al. for instance detected a global hypomethylation in tissue specimens from head and neck squamous cell carcinomas in contrast to normal mucosa [175]. Tongue squamous cell carcinoma showed a global hypomethylation as well [176]. When performing a multivariate Cox regression analysis on these samples, low 5mC methylation was significantly associated with poor disease-specific survival [176]. Ethanol-induced DNA hypomethylation can result in the expression of certain oncogenes. For instance, the oncogene Survivin (encoded by *BIRC5*) is frequently found upregulated in OSCC, probably due to promotor hypomethylation [177,178]. Survivin is an inhibitor of apoptosis. When it is expressed, cell death is inhibited which can lead to tumor progression [178]. Additionally, hypomethylation of retrotransposon elements, such as long interspersed elements (LINEs), can influence tumor formation by genome destabilization [179]. LINE-1 hypomethylation was detected in oral rinses of patients diagnosed with OSCC [180]. In accordance, analysis of LINE-1 methylation in OPMDs showed significant hypomethylation in patients where the OPMD progressed into OSCC [181]. This also translated to a worse oral cancer-free survival [181].

On the other hand, promoter hypermethylation can silence tumor suppressor genes thereby promoting carcinogenesis. In oral cancer, promoter hypermethylation of *CDKN2A*, *CDH1*, *MGMT*, and *DAPK1* has been observed [179,181,182].

#### 4.3.2. Patterns of Histone Modifications Can Change in the Presence of Ethanol

In addition to DNA methylation, also histone modifications affect transcription, DNA replication and DNA repair [164]. Various modifications are known such as histone acetylation, methylation, phosphorylation, and ubiquitination [165]. These modifications regulate the accessibility of the DNA to e.g., transcription factors. An open chromatin state, i.e., euchromatin, is associated with active gene transcription while heterochromatin is tightly packed and is therefore associated with gene silencing [165].

Ethanol similarly affects histone methylation patterns as DNA methylation discussed in the previous paragraph. Histone methyltransferases also use SAM as a methyl donor and the availability of SAM can be reduced by alcohol abuse. Consequently, histone hypomethylation occurs after chronic ethanol consumption [183]. The exact result of this is uncertain because histone methylation can lead to gene activation as well as deactivation depending on the position of the lysine residue which is modified [183].

In addition, acetylation patterns can be altered by ethanol intake. Histone acetyltransferases (HATs) catalyze histone acetylation whereby acetyl-coenzyme A (acetyl-CoA) is used as an acetyl donor. Histone acetylation typically results in euchromatin, meaning gene activation [183]. It is thought that acetate, produced during the oxidative metabolism of ethanol, can also function as a donor of an acetyl group. Acetate can be converted into acetyl-CoA by acetyl-CoA synthetase [183]. Recently, Mews et al. showed that indeed acetate coming from ethanol metabolism can lead to acetylation in the brain of ethanol-treated mice [184]. It has also been suggested that ethanol can increase histone acetylation through modification of HAT activity [185]. Another mechanism whereby ethanol can influence histone acetylation patterns might be through SIRT1 (NAD-dependent protein deacetylase sirtuin-1). SIRT1 is an important sensor that balances transcriptional activation or repression. It has an NAD^+^-dependent histone deacetylase activity [183]. Ethanol metabolism leads to a lower NAD^+^/NADH ratio which inhibits SIRT1 and this can interfere with normal acetylation patterns [183]. As the direct action of SIRT1 is histone deacetylation, more histone acetylation is expected in cells exposed to ethanol.

Mancuso et al. studied histone methylation patterns in tissue samples of patients diagnosed with OPMDs or OSCC. In comparison to healthy tissue samples, H3K4 dimethylation was increased while H3K4 trimethylation levels were decreased [186]. Interestingly, Arif et al. also reported histone 3 hyperacetylations on lysine 9 and 14 in both an OSCC cancer cell line and OSCC tissue specimens [187]. In addition, selective acetylation of histone 3 at lysine 9 (H3AcK9) has been attributed to ethanol [165,188,189]. It should be noted however this was detected in hepatic and not oral cells. These patterns of altered histone modifications are global changes that can occur throughout the genome. Both H3AcK14 and H3AcK9 have been correlated to active gene expression if these occur in coding regions [190]. If genomic locations of oncogenes are altered, this could potentially contribute to tumorigenesis. Contrarily, less H3K4 trimethylation correlates with a decreased promoter activity [191]. This can also affect oral carcinogenesis if tumor suppressor genes are hit.

## 5. Mutational Signatures Give More Insight into Carcinogenesis

Recently, studying mutational signatures in cancer exomes and genomes gained a lot of attention. Mutations represent a fingerprint of the different mutational processes that have been active over time. Therefore, studying mutational signatures can help to obtain more insight into the molecular mechanisms active during tumorigenesis [192,193]. In addition, discovery of mutational signatures might provide opportunities for new therapies. For example, Ma et al. nicely illustrated that DNA repair deficiencies result in characteristic mutational signatures. This finding can be therapeutically exploited e.g., through implementation of a synthetic lethal (personalized) cancer therapy [194].

‘Mutational signatures’ are defined as unique combinations of mutation types generated by different mutational processes. Somatic mutations are often divided into four classes: base substitutions, small indels, rearrangements, and copy number changes [192]. These can be further subclassified in biologically meaningful groups. For instance for substitutions, often the type of substitution and the sequence context are taken into account and this results in 96 possible types of mutations [192]. Multiple research groups started to extract mutational signatures from various cancer types and therefore a curated consensus of signatures was needed. The Catalogue of Somatic Mutations in Cancer or COSMIC database provides an excellent overview of the different mutational signatures already extracted [195]. Based on the class of mutations, various mutational signatures are known including single-base substitutions (SBS), doublet base substitutions (DBS), and small insertions—deletions (ID) signatures [195].

Which molecular processes underlie a specific mutational signature is an intriguing question. Somatic mutations may arise from cellular processes [196]. For instance, intrinsic APOBEC (apolipoprotein B mRNA editing enzyme, catalytic polypeptide-like) activity causes a recognizable pattern of mutations (COSMIC signatures SBS2 and SBS13) [197,198]. Additionally, exogenous and endogenous mutagens are known to cause specific signatures [196]. Signatures SBS4, DBS2, and ID3 were suggested to be associated with tobacco exposure [198]. Notably, signatures have been linked to specific environmental agents by association studies. Interestingly, a recent investigation by Kucab et al. validated these associations [199]. They sequenced human induced pluripotent stem cells after exposure to various chemicals. There was a striking similarity between the signatures extracted from these in vitro experiments and in vivo association data [199]. Taken together, this indicates that mutational signatures extracted from tumor genomes can indeed hint at the underlying mechanisms.

### Several Ethanol-Related Mutational Signatures Have Been Identified

Apart from the mutagens shortly discussed in the previous paragraph, studies have also investigated a link between ethanol consumption and the occurrence of specific mutational signatures in sequencing data derived from tumors. An overview of mutational signatures that have been associated with ethanol is given in Table 1.

As discussed in Section 4.1, acetaldehyde can accumulate in the oral cavity where it can damage the cells. Interestingly, both in vivo and in vitro studies demonstrated that acetaldehyde exposure leads to a characteristic mutational profile, dominated by GG to TT mutations [200]. This specific signature was suggested to be attributable to GG intrastrand crosslinks caused by acetaldehyde [205]. This specific signature is called DBS2 in the COSMIC database [195]. It was identified by Chen et al. in genome sequences of tobacco-related tumors, such as head and neck cell carcinoma and lung adenocarcinoma [206,207,208]. Kucab et al. afterwards confirmed that the DBS2 signature is indeed caused by acetaldehyde exposure [199].

Additionally, COSMIC signature SBS16 has been related to ethanol in various types of cancer. Chang et al. identified SBS mutational signatures in genome sequences of esophageal cancer, a cancer type with a strong ethanol etiology. The tumor samples were derived from Chinese patients only, which is important to note as this population has a high frequency of the *ALDH2*2* allele. In this study, there were six distinct SBS mutational signatures discovered of which several significantly correlated with the alcohol ingestion of the patients [201]. SBS signature E4 was found to be similar to the previously identified SBS16 signature of Alexandrov et al. [196]. Chang et al. also found a correlation between this signature and the presence of *ALDH2*2* allele, suggesting this signature is linked to acetaldehyde [201]. The results from this research were confirmed by another study in esophageal cancer samples executed by Li et al. [202]. They found a signature, resembling SBS16, which was significantly correlated with alcohol intake and the mutant *ALDH2*2* allele [202]. In liver cancer, this SBS16 signature was found as well and correlated with alcohol consumption [203]. Computational analysis of exome sequences from The Cancer Genome Atlas (TCGA) HNSCC cohort revealed that the SBS16 signature is in the top five of most commonly detected signatures [209]. Kaplan-Meier graphs also illustrated that patients with a relatively high attribution of SBS16 somatic mutations have a lower overall survival [209].

Interestingly, not all signatures identified in ethanol-related cancers show a link to acetaldehyde. Supek et al. extracted mutational signatures from TCGA data focusing on clustered mutations, defined as mutations in close proximity (≤500 base pairs) to each other [204]. Looking at these clustered mutational signatures, the authors identified a signature, C4, characterized by A > G substitutions. This signature was especially prevalent in samples from liver cancer. Signature C4 specifically associated with alcohol consumption and not with other risk factors such as smoking tobacco [204]. The authors hypothesize this ethanol-associated clustered signature arose from the translesion polymerase η, encoded by the *POLH* gene [204]. In addition to liver cancer, this C4 signature was also found in head and neck, pancreatic and esophageal cancer. Also in these types of cancer, the C4 signature was positively associated with alcohol consumption [204]. In addition, the study of Chang et al. discovered more signatures in esophageal cancer samples [201]. There was another SBS signature, called E6, correlating with alcohol drinking but not with acetaldehyde. This signature showed low similarity to any of the COSMIC signatures and therefore they believe it is new signature [201]. Currently, the underlying mechanism of this specific signature is unknown, but it might be interesting for future investigations.

## 6. Conclusions and Future Perspectives

It is clear that heavy alcohol consumption substantially increases the risk for development and malignant transformation of oral neoplasms. Multiple epidemiological association studies and animal studies established the role of ethanol in oral carcinomas and potentially malignant lesions. Despite this clear causal link, the molecular mechanisms underlying this carcinogenic effect of ethanol are still not fully revealed. The metabolites of ethanol, being acetaldehyde and reactive oxygen species, undoubtedly damage mucosal cells by the formation of DNA and protein adducts and DNA crosslinks. Additionally, ethanol is known to alter DNA and histone methylation patterns which can impact cellular survival in case key tumor suppressor genes or oncogenes are affected. However, most of this evidence was obtained from studies in the liver and the brain. An in-depth study of the epigenetic effect of ethanol in oral tissues is currently lacking which may be a proposition for further research.

Ethanol-related mutational signatures confirmed the role of acetaldehyde in esophageal and liver cancer. Nevertheless, other ethanol-related signatures were identified as well of which the mechanisms are currently not fully understood. This might indicate not all carcinogenic mechanisms of ethanol have been identified yet. Notably, there are no reports yet of mutational signatures in oral tumors or potentially malignant lesions which can be pursued in future studies. Discovery of (new) mutational signatures in oral tumors is interesting to determine or confirm underlying molecular mechanisms of ethanol-induced tumorigenesis. Additionally, mutational signatures can discover processes driving tumorigenesis such as DNA repair deficiencies. These processes can be used as a biomarker, but can also result in new therapeutic opportunities.

## Figures and Tables

**Figure 1 cancers-13-03846-f001:**
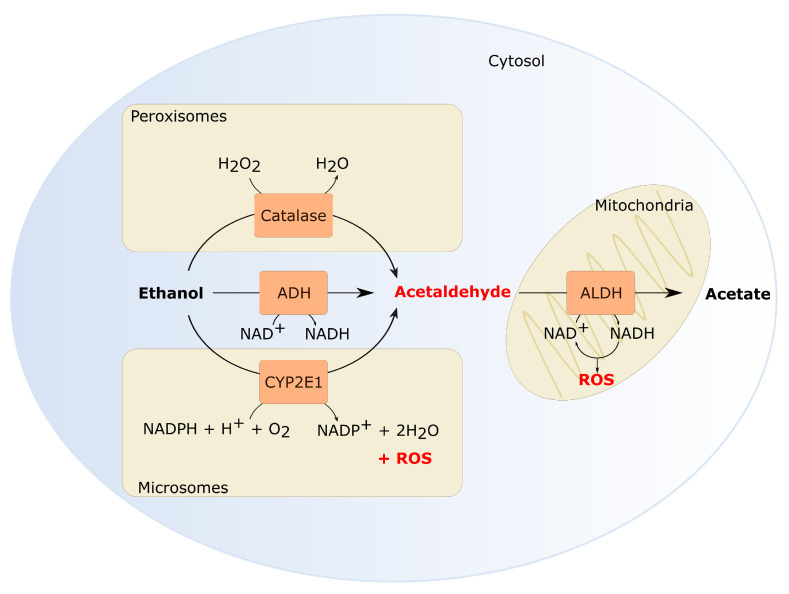
Oxidative ethanol metabolism generates reactive metabolites. Several enzymes catalyze ethanol oxidation into acetaldehyde. The cytosolic alcohol dehydrogenase (ADH) contributes the most to the formation of acetaldehyde [104]. Additionally, CYP2E1, active in the MEOS, can oxidize ethanol. Also CYP1A2 and CYP3A4 contribute to ethanol oxidation in the MEOS [105,106,107]. In normal physiological conditions, MEOS is only responsible for 25–30% of the ethanol oxidation. However, CYP2E1 activity increases significantly by heavy alcohol consumption [108,109,110]. CYP2E1 activity in the MEOS does not only generate acetaldehyde but also ROS which is another important metabolite of ethanol. Lastly, catalase can oxidize ethanol through the formation of hydroxyl radicals [111]. Catalase is especially important in the brain or in a fasted state [104,112,113]. In the next step, acetaldehyde is oxidized to acetate by aldehyde dehydrogenase (ALDH). Acetate is excreted from the cells into the systemic circulation where it can eventually be metabolized to carbon dioxide [104]. Both ADH and ALDH enzyme activity results in a decreased NAD^+^/NADH ratio. NADH reoxidation in the mitochondria can also generate ROS.

**Figure 2 cancers-13-03846-f002:**
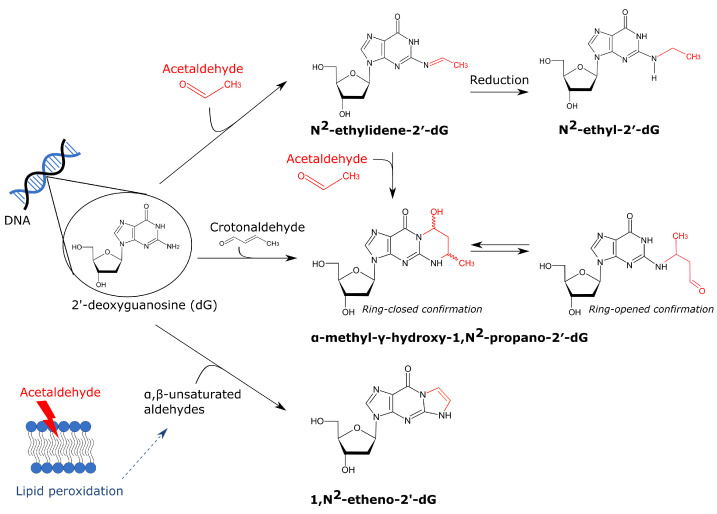
Overview of acetaldehyde-derived DNA adducts. Acetaldehyde can interact with dG which results in several adducts. The first and most abundant acetaldehyde-derived dG adduct is N^2^-ethylidene-2′-dG. It can be converted into the stable N^2^-ethyl-2′-dG by reduction by e.g., glutathione. A second adduct is α-methyl-γ-hydroxy-1,N^2^-propano-2′-dG which is formed by the interaction of either two molecules of acetaldehyde or crotonaldehyde with dG. It was suggested that in physiologically relevant conditions, two molecules of acetaldehyde are first converted into crotonaldehyde which can subsequently interact with dG [119]. This adduct can exist in a ring-closed or ring-opened confirmation. In the latter case, it can induce interstrand crosslinking (not visualized). Lastly, acetaldehyde can induce lipid peroxidation thereby forming α,β-unsaturated aldehydes. These aldehydes can also interact with dG, resulting in 1,N^2^-etheno-2′-dG.

**Figure 3 cancers-13-03846-f003:**
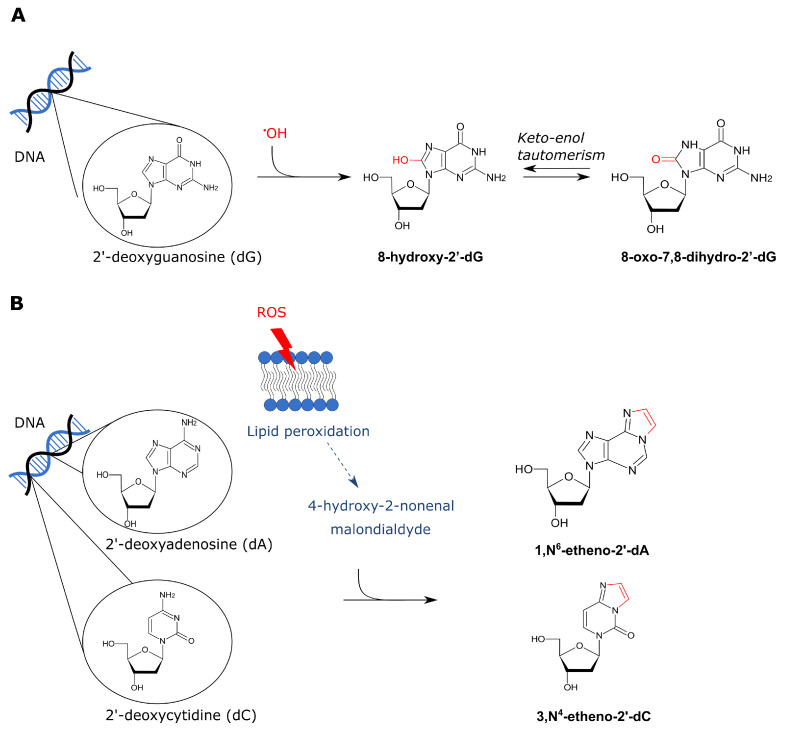
Overview of ROS-derived DNA adducts. There are both direct and indirect effects of ROS on DNA base-pairs. (**A**) Firstly, a hydroxyl radical can directly affect dG thereby forming 8-hydroxy-2′-dG. Keto-enol tautomerism favors the ketone derivative 8-oxo-7,8-dihydro-2′-dG. (**B**) On the other hand, ROS can also indirectly lead to DNA adducts via lipid peroxidation which results in formation of 4-HNE and MDA. These aldehydes can interact with DNA base pairs which results in exocyclic etheno adducts. 1,N^6^-etheno-2′-dA and 3,N^4^-etheno-2′-dC are visualized in panel B.

**Table 1 cancers-13-03846-t001:** Overview of ethanol-related mutational signatures discovered in genome sequencing data.

Signature	Cancer Subsite	Proposed Etiology	References
DBS2	Lung, Head and Neck	Acetaldehyde exposure	[199,200]
SBS16	Esophagus, Liver	Acetaldehyde exposure	[201,202,203]
C4	Liver, Head and Neck, Esophagus, Pancreas	Translesion polymerase η	[204]
E6	Esophagus	Unknown	[201]

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
