# Peer review of "Ethanol-Induced Cell Damage Can Result in the Development of Oral Tumors"

_cancers, 2021, doi:10.3390/cancers13153846_

Round 1

Reviewer 1 Report

This review restates the points made in the Am Cancer Assoc and Nat Inst of Health guidelines.  However, there is some bias reflected in the lack of emphasis on ethanol consumption dose and duration as a risk factor for oral and other) cancer.

There is sufficient evidence that low-moderate ETOH consumption may have beneficial cardiovascular effects, which the authors chose to ignore. The contribution of low ETOH consumption to oral cancer is controversial.  Additionally, there is no robust evidence that ETOH is a risk factor for lip cancer, as the authors claim.  These claims need to be moderated due to their controversial status.

The history of ETOH beginnings in ancient times is irrelevant and superfluous, and should be deleted.  

The correlation of smokeless tobacco alone with oral cancer is also controversial.  Suggest reviewing Rodu et al.

Lastly, I'd suggest reviewing the ETOH effects on DNA repair mechanism.

Author Response

Reviewer #1 (Remarks to the Author):
This review restates the points made in the Am Cancer Assoc and Nat Inst of Health guidelines.
However, there is some bias reflected in the lack of emphasis on ethanol consumption dose and duration as a risk factor for oral and other) cancer. There is sufficient evidence that low-moderate ETOH consumption may have beneficial cardiovascular effects, which the authors chose to ignore.

We agree with reviewer #1 that our review does restate governmental guidelines that strongly recommend moderation of alcohol consumption. However, a recent study (2021) by Calvart et al published in BMC public Health highlighted that public awareness of the role of ethanol in mouth and throat cancer was only
66% in the study group. Also Rumgay et al. (2021, The Lancet Oncology) stated that efforts to increase public awareness regarding the role of alcohol in cancer are needed to diminish alcohol-related cancer cases. Therefore, we believe that publishing our review, restating the role of ethanol in oral tumorigenesis,
is valuable.

There is indeed compelling evidence that low alcohol consumption is cardio protective, as visualized in Griswold et al. (2018, The Lancet Oncology). Nevertheless, the scope of our review paper was not to discuss
all of the known alcohol-associated effects on human health. The emphasis of this work is on ethanol as a risk factor for oral carcinogenesis. Taken into account the remark of reviewer #1, we added this information and clarified the scope of the review (Line 61 – 66). 

The contribution of low ETOH consumption to oral cancer is controversial.
Additionally, there is no robust evidence that ETOH is a risk factor for lip cancer, as the authors claim. These claims need to be moderated due to their controversial status.

Multiple recent meta-analyses have estimated that even low alcohol intake, usually defined as ≤ 12.5 g ethanol, is associated with an increased risk of developing oral tumors (e.g. Turati, 2010, Oral Oncology; Bagnardi, 2013, Annals of Oncology; Griswold, 2018, The Lancet; Di Credico, 2020, British Journal of
Cancer). Nevertheless, some individual studies showed contrasting results regarding the impact of low ethanol consumption. We added this nuance to our manuscript on line 150 - 155.

With regards to lip cancer, we simply restated the findings from other publications and studies. In the cited research, there is mostly no distinction made between lip cancers and other cancer types from the oral cavity tissue. Also in the WHO global status report on alcohol and health, ‘lip and oral cavity cancers’ are always grouped. To be as accurate as possible, we always used exactly the same definitions as in the papers cited. We believe this is the best possible way to cite research which is not our own, and therefore we did not adapt our manuscript. Below are direct citations from the publications we refer to in our manuscript. 

 Abstract: World Health Organization, 2018, Global status report on alcohol and health 2018
On page 70 of this report, it is stated that “Furthermore, alcohol had the largest impact on cancers of the upper aerodigestive tract, being responsible for 26.4% of all lip and oral cavity cancers, 30.5% of all other pharyngeal cancers (excluding nasopharyngeal cancers), 21.6% of all laryngeal cancers, and 16.9% of all oesophageal cancers.”
 Line 75 : Bray et al., 2018, CA: A Cancer Journal for Clinicians In Table 1 it is stated that in 2018, 354,864 of new cases of lip and oral cavity cancer are reported.
 Line 135: Griswold et al., 2018, The Lancet Figure 4 of this publication shows “Relative risk curves for selected conditions by number of standard drinks consumed daily. B) Relative risk curves for lip and oral cavity cancer, ischaemic
heart disease, diabetes, and tuberculosis for males.”
 Line 202: Soffritti et al., 2002, Annals of the New York Academy of Sciences
As stated in their abstract “Treatment started at 39 weeks of age (breeders), 7 days before mating, or from embryo life (offspring) and lasted until their spontaneous death. Under tested experimental conditions, methyl alcohol and ethyl alcohol were demonstrated to be carcinogenic for various organs and tissues. They must also be considered multipotential carcinogenic agents.
In addition to causing other tumors, ethyl alcohol induced malignant tumors of the oral cavity, tongue, and lips.” 

The history of ETOH beginnings in ancient times is irrelevant and superfluous, and should be deleted.
It is true that this information is not crucial for our review paper. However, we still believe this is a nice addition since it is remarkable that the adverse effects of ethanol are already known for so long while the broad public awareness is relatively recent. Because reviewer #2 did enjoy reading the brief history of
alcohol consumption and it’s adverse effects, we chose to only shorten this section without completely removing it (Line 29 – 43). 

The correlation of smokeless tobacco alone with oral cancer is also controversial. Suggest reviewing Rodu et al.
The monograph vol. 89 of the International Agency for Research on Cancer (entitled ‘Smokeless Tobacco and Some Tobacco-specific N-Nitrosamines’) published in 2007 contains a variety of studies which showed that smokeless tobacco is independently associated with oral cancer. Also, the study cited in our
manuscript (Siddiqi et al., 2020, BMC Medicine) corrected for smoking tobacco as confounding factor and still found a significant effect of smokeless tobacco on the risk for oral tumors. When revising the publication of Siddiqi et al., we however noticed that in North-European and North-American studies,
smokeless tobacco did not significantly increase the risk for oral tumors. We added this nuance as an update to our manuscript (Line 90 – 95).

We evaluated Rodu et al. (2019, Harm Reduction Journal) and observed that this study only quantifies mortality in male users of tobacco (both smoked or smokeless). Indeed, Rodu et al. found no significant increase of mortality from malignant oral neoplasms and the use of smokeless tobacco alone. However,
this does not necessarily mean that the incidence or risk of developing oral tumors is not correlated with smokeless tobacco.  

Lastly, I'd suggest reviewing the ETOH effects on DNA repair mechanism

In the previous version of our manuscript, we discussed several DNA repair mechanisms for ethanolinduced DNA damage in sections 4.2.1 and 4.2.2. The metabolites of ethanol, acetaldehyde and ROS, can introduce mutations ad random in the genome of eukaryotic cells. It is possible that key genes of the DNAdamage response are affected by this, but this is not a given. Therefore, we could not discuss the effect of ethanol on specific DNA repair pathways. We hope this clarification is sufficient for this comment of reviewer #1. 

Reviewer 2 Report

In this review titled "Cell damage due to ethanol exposure is related to oral carcinoma" the authors have performed an excellent job putting together a vast issue into one integral story. Lore et al. especially did a good job reviewing a brief history of alcohol consumption and discussed the "biochemical" and "molecular" impact on oral carcinogenesis. The reviewer considers the current manuscript a "useful progress" over previous reviews that discussed the effect of alcohol on oral carcinogenesis. 

The content of this review is appropriate, reasonably precise, and scientifically correct.

This reviewer has a few minor concerns, especially on "relatively less formal approach of writing flow," and recommends further consideration in rephrasing/modifications to make the manuscript a better read

  1. Although the title of this manuscript contains appropriate keywords [facilitate future web search], this reviewer is requesting a change to make it more direct.
  2. Line. 21-28- "During the following centuries---" This section has readability concerns and sentence continuity issues. Sentence rephrasing might make it more readable.
  3. Line 42 "Nowadays, it is crystal clear that heavy alcohol"   might be rephrased as "Evidently, alcohol consumption negatively impacts human health."
  4. 62-  why Oral cancers are more prevalent in men? Add brief reasoning.
  5. "In 2018, 354,864 new patients were diagnosed with lip or oral cavity cancers"- the denominator is missing.
  6. Line 64: Adding a few examples of the mid-income countries will make this line a more appropriate read.
  7. Further reference is needed to support "Tobacco and alcohol have a major synergistic effect on the development of cancerous lesions with odds ratios increasing up to 170.
  8. "Interestingly, no effect of duration of drinking was observed for oral cancers"- Please adopt a direct sentence and mention which aspect of oral cancer has "no effect" [from the duration of drinking]
  9. Please expand OPMD in line 130
  10. Line 124-126: Please consider writing a direct/straightforward sentence
  11. all "Lichen planus" should be "oral lichen planus".
  12. The manuscript should include specific examples citing acetaldehyde-driven DNA hypomethylation/Histone acetylation/histone demethylation leading to transcriptional activation of relevant oncogenes etc. Specific examples with reference will be ideal in sections 4.3.1 and 4.3.2
  13. DNA-Protein adduct and associated inflammation: Consider discussing the significance.
  14. The conclusion and future perspectives section lack a section suggesting potential "significance/application of ethanol exposure mutational signatures" in oral cancer diagnostics/therapy decision etc.

Author Response

Reviewer #2 (Remarks to the Author):
In this review titled "Cell damage due to ethanol exposure is related to oral carcinoma" the authors have performed an excellent job putting together a vast issue into one integral story. Lore et al. especially did a good job reviewing a brief history of alcohol consumption and discussed the "biochemical" and
"molecular" impact on oral carcinogenesis. The reviewer considers the current manuscript a "useful progress" over previous reviews that discussed the effect of alcohol on oral carcinogenesis.

The content of this review is appropriate, reasonably precise, and scientifically correct.

This reviewer has a few minor concerns, especially on "relatively less formal approach of writing flow," and recommends further consideration in rephrasing/modifications to make the manuscript a better read

1. Although the title of this manuscript contains appropriate keywords [facilitate future web search], this reviewer is requesting a change to make it more direct.
We agree with reviewer #2 that a more direct title is better. The title of our manuscript was adapted accordingly as “Ethanol-induced cell damage can result in the development of oral tumors”.
2. Line. 21-28- "During the following centuries---" This section has readability concerns and sentence continuity issues. Sentence rephrasing might make it more readable.
We decided to rephrase and also shorten this paragraph to also meet the comments of reviewer # 1. The lines which reviewer #2 is referring to were deleted in the updated text. The rephrased paragraph can be found on line 29 - 43.
3. Line 42 "Nowadays, it is crystal clear that heavy alcohol" might be rephrased as "Evidently, alcohol consumption negatively impacts human health."
This was adapted in the revised manuscript (Line 45).
4. 62- why Oral cancers are more prevalent in men? Add brief reasoning.
Generally, it is believed that men are more exposed to alcohol and tobacco, which are the primary risk factors for developing oral cancers. For instance, the study of Kanny et al. (2018 in American journal of preventive medicine) observed that “The prevalence of binge drinking among men (22.2%) was about twice that of women (12.1%, p<0.0001)”. As the risk for developing oral tumors increases
linearly with the dose of ethanol that is consumed, men are more at risk for these cancers. We included a brief statement in our manuscript (Line 73 – 74 & 102 - 103).

5. "In 2018, 354,864 new patients were diagnosed with lip or oral cavity cancers"- the denominator is missing.
In 2018, around 18.1 million new cancer cases were reported. From these, around 350,000 cases were lip and oral cavity cancers which roughly corresponds to 2% of all newly diagnosed patients.
The latter was added to the manuscript (Line 75 – 76).
6. Line 64: Adding a few examples of the mid-income countries will make this line a more appropriate read.
We included three countries in this sentence. It know reads as “Many of these cases occur in lowand middle-income countries, e.g. India, Pasistan, or Tanzania, making it the fourth most common type of cancer in these countries” (Line 77).
7. Further reference is needed to support "Tobacco and alcohol have a major synergistic effect on the development of cancerous lesions with odds ratios increasing up to 170.
We removed our statement that odds ratios could increase up to 170. This statement was based on the publication of Dal Maso et al. (2016 in European Journal of Epidemiology). They used a bidimensional logistic spline model to investigate the dose-response relationship between alcohol intake and smoking tobacco. An odds ratio of 172.25 was obtained upon drinking 160-180 g of pure
ethanol a day in combination with smoking 35-40 cigarettes a day.
We agree with reviewer #2 that this is not a realistic example. We included more references pointing out a synergistic effect of tobacco and alcohol and we added a nuance in our text that the increase of the odds ratio is probably highly dependent on the dose of both alcohol and tobacco (Line 96 - 98).
8. "Interestingly, no effect of duration of drinking was observed for oral cancers"- Please adopt a direct sentence and mention which aspect of oral cancer has "no effect" [from the duration of drinking]
We adapted this in our current manuscript (Line 142 – 145).
9. Please expand OPMD in line 130
We expanded this abbreviation at line 82 which is the first time we mention OPMDs in our text.
10. Line 124-126: Please consider writing a direct/straightforward sentence
Indeed, this was a weirdly structured sentence. We adapted this in our revised manuscript (Line 146
– 147).
11. all "Lichen planus" should be "oral lichen planus".
We adapted this in our revised manuscript.
12. The manuscript should include specific examples citing acetaldehyde-driven DNA hypomethylation/Histone acetylation/histone demethylation leading to transcriptional activation of relevant oncogenes etc. Specific examples with reference will be ideal in sections 4.3.1 and 4.3.2
We agree with reviewer #2 that more concrete examples are interesting to add to our review. Regarding altered DNA methylation patters, we already referred to LINE-1 which is often hypomethylated in OSCC. This is a retrotransposon element which is in normal cells inactivated by DNA methylation. If this element gets activated, e.g. through loss of methylation, it can destabilize the genome which could result in tumor formation. We added an extra example, i.e. Survivin. This is an inhibitor of apoptosis which can be activated through promoter hypomethylation. If expressed, it leads to dysregulated cell death which is a hallmark of cancer cells (Line 434 - 437).
We acknowledge that these examples are indirect because there are no reports stating that ethanol exposure leads to hypomethylation of exactly these genomic regions. Nevertheless, we believe that ethanol can alter DNA methylation patterns randomly, and if key genes are hit this could lead to the
development or progression of a cancer cell.

Regarding the histone modifications, the effect of ethanol on gene expression is less straightforward. Whether or not a histone modification results in gene activation or repression highly depends on the amino acid which is modified and the modification itself (i.e. methylation or acetylation). We searched through literature, but unfortunately could not find an example of a specific gene which
activity is up- or downregulated due to ethanol-induced histone alterations. The only thing we can state for sure is whether the modifications discussed in our review are related with gene activation or repression. This could contribute to carcinogenesis if oncogenes or tumor suppressor genes are affected respectively (Line 482 – 488).
13. DNA-Protein adduct and associated inflammation: Consider discussing the significance.
Tissue inflammation due to antibody cell dependent cytotoxicity has mostly been studied in liver tissues. Nevertheless, acetaldehyde and ROS-derived protein adducts were also observed in oral tissues. Additionally, tissue inflammation has also been linked to the development of oral tumors. We
added a sentence to clarify the significance in our manuscript (Line 369 – 370).
14. The conclusion and future perspectives section lack a section suggesting potential "significance/application of ethanol exposure mutational signatures" in oral cancer diagnostics/therapy decision etc.
This was added to the revised text (Line 494 – 497 & 597 – 601). 

Round 2

Reviewer 1 Report

The authors have responded adequately to most criticisms and introduced appropriate modifications to their review paper.  Nevertheless, some areas of inadequacy remain.

This reviewer believes that scientific literature must be data-driven.  Parroting previous reviews /opinion pieces / monographs without applying scientific critical thinking simply helps in continuing propagation of misinformation.  Lip cancer is associated with UV exposure and there are no robust data supporting any role for ETOH.  The fact that the authors persist in this error simply because others have done so is disappointing.

Some of the epidemiologic data in the paper are dated.  For example, the 75% of oral cancers associated with ETOH and tobacco is taken from a 2005 paper.  These data have changed dramatically, with about 60% of these lesions attributed to HPV in North America in 2020.

Similarly, the authors fail to mention the association of oral cancer with betel nut chewing, which may be responsible for a majority of lesions in Asia and Africa.

Lastly, I continue to believe that both the intro and the conclusion are long-winded and superfluous.  The ancient history of alcohol may be interesting and fun, but that does not make it science.  The Conclusion restates material covered in discussion.

The authors need to understand some of the fundamental requirements of good science, which starts with unbiased data and ends with critical thinking.  While hard work is appreciated, it is hardly sufficient.

Author Response

Rebuttal reviewer #1

The authors have responded adequately to most criticisms and introduced appropriate modifications to their review paper. Nevertheless, some areas of inadequacy remain.

This reviewer believes that scientific literature must be data-driven. Parroting previous reviews /opinion pieces / monographs without applying scientific critical thinking simply helps in continuing propagation of misinformation. Lip cancer is associated with UV exposure and there are no robust data supporting any role for ETOH. The fact that the authors persist in this error simply because others have done so is disappointing.

We added a nuance in our manuscript (Line 139-141).

Some of the epidemiologic data in the paper are dated. For example, the 75% of oral cancers associated with ETOH and tobacco is taken from a 2005 paper. These data have changed dramatically, with about 60% of these lesions attributed to HPV in North America in 2020.

We disagree with the statement of reviewer 1 that 60% of the oral lesions are attributed to HPV. HPV is especially associated with oropharyngeal cancers. Indeed, in oropharyngeal cancers HPV-positive lesions are prevalent and the incidence is increasing. Prof. Nuyts has 13 peer-reviewed publications on this topic specifically.

However, in our review we are focusing on oral cavity cancer (OCSSC). Estimates on HPV positivity in OCSCC vary between geographical regions, e.g. 36% in Japan and only 2% in the Netherlands. We added this to our manuscript (Line 106-108).

Similarly, the authors fail to mention the association of oral cancer with betel nut chewing, which may be responsible for a majority of lesions in Asia and Africa.

The scope of our review is how ethanol-induced damage can contribute to the development of oral lesions. Our review is not an overview of all the known risk factors for oral cavity cancers. On this latter topic, there exist multiple textbooks and reviews. The goal of the second paragraph in section 2 was primarily to highlight the fact that heavy alcohol consumption is often accompanied by smoking tobacco and that, because of a synergistic effect, it is not always straightforward to disentangle the contributions of both individual risk factors.

Lastly, I continue to believe that both the intro and the conclusion are long-winded and superfluous. The ancient history of alcohol may be interesting and fun, but that does not make it science. The Conclusion restates material covered in discussion.

In the previous round of review, we already shortened the introduction (section 1). We believe a review is a summary of existing scientific information, including old scientific information (e.g. the work of prof. R. Pearl on alcohol and longevity).

The primary goal of a conclusion is to repeat the ‘take-home messages’ of the paper, which is exactly why we restate some findings that were covered in the other sections of our review. We did however shorten the conclusion (Line 579-596).

The authors need to understand some of the fundamental requirements of good science, which starts with unbiased data and ends with critical thinking. While hard work is appreciated, it is hardly sufficient

As senior researcher and expert clinician with more than 150 peer-reviewed Pubmed publications and presentations at numerous international meetings, prof. Nuyts knows what good science is.
